# Accelerating Structured Chain-of-Thought in Autonomous Vehicles

## Abstract

Chain-of-Thought (CoT) reasoning enhances the decision-making capabilities of vision-language-action models in autonomous driving, but its autoregressive nature introduces significant inference latency, making it impractical for real-time applications. To address this, we introduce FastDriveCoT, a novel parallel decoding method that accelerates template-structured CoT. Our approach decomposes the reasoning process into a dependency graph of distinct sub-tasks, such as identifying critical objects and summarizing traffic rules, some of which can be generated in parallel. By generating multiple independent reasoning steps concurrently within a single forward pass, we significantly reduce the number of sequential computations. Experiments demonstrate a 3-4× speedup in CoT generation and a substantial reduction in end-to-end latency across various model architectures, all while preserving the original downstream task improvements brought by incorporating CoT reasoning.

## 1 Introduction

In recent years, language has emerged as a key modality in robotic systems, enabling progress in domains such as manipulation (Driess et al., 2023; Zitkovich et al., 2023) and autonomous driving (Huang et al., 2024; Tian et al., 2024b). With the rapid advances in Large Language Models (LLMs) and Vision–Language Models (VLMs), language has increasingly been integrated into perception and decision-making pipelines, transforming Visual–Action (VA) models into Vision–Language–Action (VLA) models (Black et al., 2024; Kim et al., 2024). Compared to traditional VA models, VLAs benefit from language grounding, which enhances their ability to interpret user intent, decompose tasks, and apply common-sense reasoning for more human-like behavior.

A representative technique in this direction is Chain-of-Thought (CoT) prompting (Wei et al., 2022), which sacrifices some inference efficiency in exchange for improved reasoning accuracy. By encouraging VLAs to break down complex problems into a sequence of simpler subproblems, CoT leverages inference-time scaling to improve policy performance. Beyond prompting, CoT has also been incorporated into training pipelines, for example through curated CoT datasets in supervised fine-tuning (SFT) (Cui et al., 2025; Tian et al., 2024a). Following the release of DeepSeek v3 (Liu et al., 2024), reasoning with extended *structured* CoT traces has gained significant traction in the robotics community, with natural extensions into autonomous vehicles (AVs) (Zhao et al., 2025).

Autonomous driving differs from pure language tasks or manipulation in its strict requirement for inference speed. In typical AV policies, decisions must be updated at a high frequency (often 10 Hz or more) to safely respond to rapidly changing environments, which imposes strict constraints on the number of tokens that can be generated within each planning cycle. However, a standard CoT trace often includes multiple stages (e.g., environment description, identification of critical objects, meta-action prediction) which add up to hundreds of additional tokens and a significant inference overhead, making its application to AV challenging.

While the sequential nature of autoregressive decoding makes CoT reasoning a bottleneck during inference, reasoning in AVs comprises multiple components that are largely *independent*, and thus amenable to parallelization. Much like a human driver, an AV agent can assess environmental factors such as road conditions, traffic signs, and critical objects in parallel. This characteristic of AV scenarios makes them particularly well-suited for a high degree of parallelism, in contrast to recent works (Jin et al., 2025; Yang et al., 2025b; Pan et al., 2025) focused on general reasoning tasks

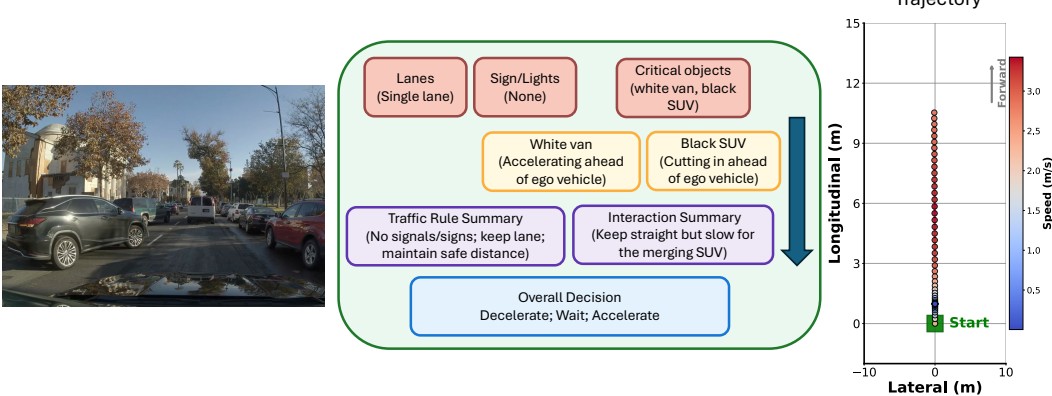

Figure 1: Introducing a CoT template can help improve the accuracy of meta-action prediction and trajectory generation in AVs. Further structuring CoT with a template containing specific topic dependencies makes CoT decoding parallelizable.

(e.g., mathematics) where the number of decomposable independent sub-tasks is often limited and problem-specific. Furthermore, AV agents typically follow a structured reasoning pattern based on a fixed set of observational factors to ensure safe driving. This regularity eliminates the need for the ad-hoc task decomposition required in other domains and enables the use of a more sophisticated algorithm to orchestrate parallel generation of CoT traces.

**Contributions.** Based on this intuition, we propose **FastDriveCoT**, a method for accelerating CoT inference in AV tasks. Our approach introduces a systematic way to format CoT traces into a structured template. To maximize the degree of parallelism, we use an optimal algorithm that dynamically identifies which fields can be generated concurrently based on a general dependency graph. Furthermore, we optimize the LLM inference process for maximum efficiency with our parallel decoding strategy, with a particular focus on how fields are arranged and merged. Our experiments show that FastDriveCoT significantly accelerates CoT generation, achieving a 3.1 to 4.1× speedup over autoregressive decoding while preserving the downstream task improvements brought by incorporating CoT reasoning. We further validate that these findings hold in many settings, across meta-action and trajectory prediction tasks in both autoregressive and transfusion (Zhou et al., 2025a) frameworks.

## 2 RELATED WORKS

### 2.1 VLA MODELS FOR AUTONOMOUS VEHICLES

Large language models (LLMs) exhibit broad world knowledge and strong in-context learning, making them promising for tackling long-tail scenarios in autonomous driving that conventional perception–planning pipelines struggle to handle. Recent work brings these capabilities into planning through VLA models. DriveVLM (Tian et al., 2024b) proposes a VLM-centric stack for scene description, analysis, and hierarchical planning. To mitigate computational latency and address spatial-reasoning gaps, it adopts a dual-system design in which a slow VLM performs high-level planning while a fast expert policy handles real-time control, demonstrating improvements on the nuScenes benchmark and in on-vehicle tests. In parallel, end-to-end VLA approaches map multimodal inputs directly to actions. (Xu et al., 2024; Renz et al., 2024) use VLMs that take in 2D videos features and are jointly instruction-tuned for scene-centric language outputs and action prediction. (Zhou et al., 2025b) augments inputs with 3D cues, such as birds-eye-view (BEV) features and structured perception vectors, providing richer spatial context that improves 3D understanding and action prediction. However, these approaches have yet to fully exploit the language capabilities of LLMs for inference-time reasoning to improve action prediction performance.

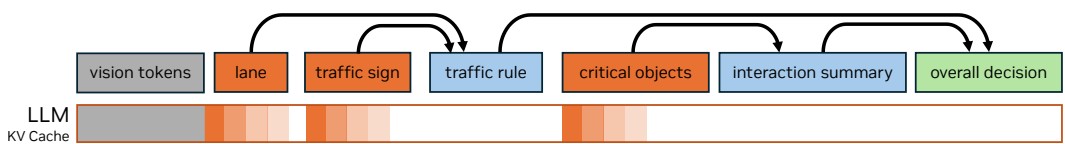

Figure 2: Parallel decoding of structured chain-of-thought (CoT) in a VLA. The CoT is decomposed into a template of fields with certain dependencies, represented by edges. All independent fields, whose dependecies have been satisfied, are decoded at the same time, such as lane, traffic sign, and critical objects as shown in the figure. Background shading within the LLM shows the updating of the KV cache.

## 2.2 CHAIN-OF-THOUGHT IN AUTONOMOUS VEHICLES

CoT prompting (Wei et al., 2022) improves multi-step reasoning by eliciting intermediate rationales, and recent reasoning-first systems (OpenAI; Comanici et al., 2025; Guo et al., 2025) show that allowing models to spend more time thinking reliably boosts accuracy. In autonomous driving, CoT has been integrated into VLA pipelines. (Wang et al., 2024; Li et al., 2024) release end-to-end datasets with explicit reasoning traces. (Renz et al., 2025) generates a commentary about what to do and why before predicting actions, achieving state-of-the-art performance on the CARLA Leaderboard 2.0 (CARLA Simulator Team, 2023) and Bench2Drive (Jia et al., 2024). AutoVLA (Zhou et al., 2025c) introduces fast/slow reasoning with GRPO fine-tuning to invoke long reasoning only when necessary. AutoDrive-$R^2$ (Yuan et al., 2025) aligns reasoning with trajectories to further improve action prediction performance. Despite these advances, inference latency remains a bottleneck. Our approach leverages parallel decoding to accelerate CoT reasoning *without* sacrificing action prediction performance.

## 2.3 TASK DECOMPOSITION AND PARALLEL DECODING IN LLMS

Several lines of research in LLM reasoning are relevant to our work. One prominent area is task decomposition, where methods like Least-to-Most prompting (Zhou et al., 2022), Tree of Thoughts (ToT) (Yao et al., 2023), and Reasoning-via-Planning (RAP) (Hao et al., 2023) break complex problems into simpler sub-tasks. Similarly, parallel decoding techniques such as Self-Consistency (Wang et al., 2022) and Best-of-N (Brown et al., 2024) generate multiple reasoning traces concurrently to improve final answer quality through voting. However, both of these research directions primarily focus on enhancing task performance, often at the cost of increased latency.

In contrast, some other works prioritize efficiency. One such line of research accelerates decoding at the token level through methods like speculative (Leviathan et al., 2023) and lookahead decoding (Fu et al., 2024), though their speedup is fundamentally limited by the accuracy of the speculative predictions. More closely related to our approach is a recent line of work on independent sub-task parallelism, including methods like Hogwild (Rodionov et al., 2025), Pasta (Jin et al., 2025), APR (Pan et al., 2025), and Multiverse (Yang et al., 2025b). While these methods aim for efficiency, their speedup is often constrained by the limited number of parallelizable branches in general reasoning tasks and the computational overhead required to first analyze and decompose the problem.

## 3 METHODS

Standard autoregressive generation in LLMs is memory-bound on modern GPUs (due to KV-cache operations) and thus not able to fully saturate the hardware's compute capabilities with a single token per forward pass (Kwon et al., 2023; Dao et al., 2022; Dao, 2023). This underutilization of the GPU creates an opportunity for parallel decoding, a technique that generates multiple tokens simultaneously to improve efficiency.

While prior works have explored sub-task decomposition to enable parallel decoding (Yang et al., 2025b; Jin et al., 2025; Rodionov et al., 2025), they have largely focused on general reasoning domains such as mathematics and coding. These domains typically require complex, task-specific decomposition strategies and inherently offer a limited degree of parallelism, resulting in limited speedup. In contrast, AV tasks frequently follow a standardized CoT pattern, progressing from

environment descriptions to critical object identification and finally to meta-action predictions. This structured reasoning process is naturally suited for a detailed sub-task decomposition, enabling CoT generation with a high degree of parallelism.

To improve inference efficiency for AV tasks, we introduce FastDriveCoT, a parallel decoding method that leverages the structured reasoning process described above. We first introduce a CoT template tailored for the above standardized CoT pattern of AV tasks (Section 3.1). To manage and maximize parallelism, we develop a dependency graph and dynamic programming algorithm that adapts to varying field lengths (Section 3.2). Finally, we detail our design for handling attention masks, position IDs, padding, and the KV-cache. This implementation is designed with consideration for critical bottlenecks in LLM inference to ensure maximum performance (Section 3.3). An overview of FastDriveCoT is shown in Figure 2.

## 3.1 TEMPLATE COT

Our CoT template decomposes the AV reasoning task into a sequence of specific fields. The initial fields are designed to capture the driving environment and key entities within it, including *lighting*, *road condition*, *weather*, *type of junction*, *type of road*, *lanes*, *critical objects*, *traffic light*, *traffic sign*, and *additional traffic regulation*. The *critical objects* field, in particular, is intended to encompass vehicles, pedestrians, cyclists, obstacles, or any other objects relevant to driving safety. For each *critical object*, we additionally include its *relative position*, *object type*, and *justification*. The next set of fields provide structured summarization of the scene, covering *traffic regulation*, as well as *non-interactive* and *interactive* elements. Finally, the template concludes with an overall summary of the expected *ego behavior*. Each of these fields is designed to be populated with free-form natural language of varying length. The particular chain-of-thought template presented here is intended as an example; in practice, such templates can be adjusted depending on the application and system setup.

Certain fields, such as *lanes* and *critical objects*, can contain a variable number of instances. For example, the lane configuration changes as the vehicle proceeds, and the number of critical objects varies from case to case. A naive approach of describing all instances within a single free-form field would yield a slow, sequential generation process. To enable parallelism, we introduce a two-stage process for these multi-instance fields.

- Stage 1 (Enumeration): The model first generates a high-level overview. For lanes, this involves identifying distinct time ranges for analysis. For critical objects, this means enumerating each individual object to be described.
- Stage 2 (Elaboration): The model then elaborates on the details for each instance identified in the first stage.

This structure allows the detailed descriptions for multiple lane time ranges or multiple critical objects to be generated in parallel, significantly improving inference efficiency. To simplify the implementation, we define a fixed number of slots for these multi-instance fields. Our template allocates space for 3 time ranges for *lanes* and 4 *critical objects*, which correspond to the maximum counts for each respective category observed in the training data. In cases where the actual number of instances is less than the allocated amount, the remaining slots are populated with a "N/A" placeholder. A more adaptive approach would be to dynamically adjust the template based on the enumeration results from the first stage, which we leave as a direction for future work.

The fields are formatted with one entry per line using the structure "*field name: field content*". These lines are then concatenated to form the complete CoT text, which typically results in a total length of 300 to 500 tokens.

## 3.2 DEPENDENCY GRAPH

The fields within our CoT template exhibit a mix of dependencies. Some fields are mutually independent, such as *weather* and *road condition*, and can therefore be decoded in parallel. Other fields, however, have dependencies that dictate a specific generation order. For instance, the *summary of traffic rules* cannot be generated until the *traffic signs* and *traffic lights* fields are complete. Similarly, for multi-instance fields like *lanes* and *critical objects*, the *enumeration* stage must

precede the *elaboration* stage. This required generation order must be strictly enforced during inference.

Managing these dependencies requires a general method that can adapt to various relationship structures. This challenge is compounded by length variability: the fields in the template have different lengths from one another, and the same field may have a different length across different cases. This unpredictability makes a fixed decoding schedule impractical. To address these challenges and optimize performance, we introduce a **dependency graph**. This data structure allows our system to dynamically track the generation process and, at any step, identify which fields have their prerequisites satisfied and are ready for parallel decoding.

---

**Algorithm 1** Parallel CoT decoding using dependency graph

---

**Require:** Dependency graph $\mathcal{G}$
1:  $d_v \leftarrow$ the number of incoming edges of $v, \forall v \in V(\mathcal{G})$
2:  $S \leftarrow \{v | d_v = 0\}$
3:  **while** $S \neq \varnothing$ **do**
4:      Decode a new token in each $v \in S$ in parallel
5:      **for** $v \in S$ **do**
6:          **if** the field corresponding to $v$ is finished **then**
7:              Remove $v$ from $S$
8:              **for** $u : (v, u) \in E(\mathcal{G})$ **do**
9:                  $d_u \leftarrow d_u - 1$
10:                 **if** $d_u = 0$ **then**
11:                     Add $u$ to $S$
12:                 **end if**
13:             **end for**
14:         **end if**
15:     **end for**
16: **end while**

---

The dependency graph is a directed acyclic graph (DAG) where each node represents a field in the template. A directed edge from the node for field $A$ to the node for field $B$ indicates that the generation of field $B$ is directly dependent on the prior completion of field $A$. Figure 3 shows an example of a simplified dependency graph.

We use a dynamic programming (DP) algorithm to schedule the parallel decoding based on the dependency graph. First, an initialization step is performed: the set of "ready" fields is populated with all source nodes (i.e., those without any incoming edges). The algorithm then proceeds iteratively until all fields are fully generated. In each step of the iteration:

- A token is generated in parallel, in a single forward pass of the LLM, for every field in the ready set.
- When a field's generation is complete, its node signals to all dependent nodes.
- A node is added to the ready set as soon as it has received signals from all of its prerequisite nodes.

The pseudocode for this algorithm is provided in Algorithm 1.

Our scheduling algorithm is optimal with respect to the number of forward passes, as it completes the generation using the minimum number possible. This minimum is equal to the length of the critical path, which is defined as the maximum cumulative number of tokens along any dependency chain in the graph. As we will later demonstrate in the experiments, this optimality in minimizing forward passes directly translates to maximizing the scheduling algorithm's speedup.

### 3.3 LANGUAGE MODEL INFERENCE

Generating multiple fields in parallel necessitates an efficient method for managing and combining their outputs. Many previous works that decompose reasoning into sub-tasks simply generate each part independently and then concatenate the resulting natural language text (Zhou et al., 2022; Yao et al., 2023; Hao et al., 2023). However, this approach is computationally inefficient as the KV cache cannot be shared or reused across the parallel generation of different fields, leading to redundant memory operations and computations.

To overcome this inefficiency, we propose formatting the entire template CoT as a single, continuous sequence. When decoding tokens for multiple fields in parallel, we pack them all together along the sequence length dimension. This approach allows the model to compute next-token logits for all

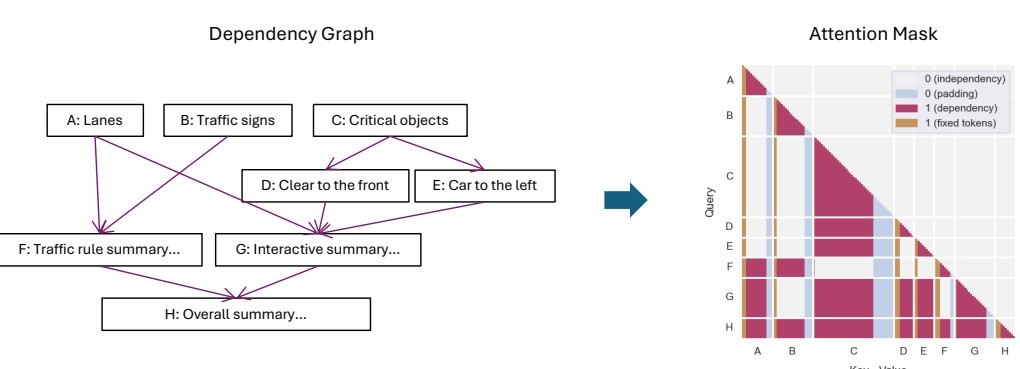

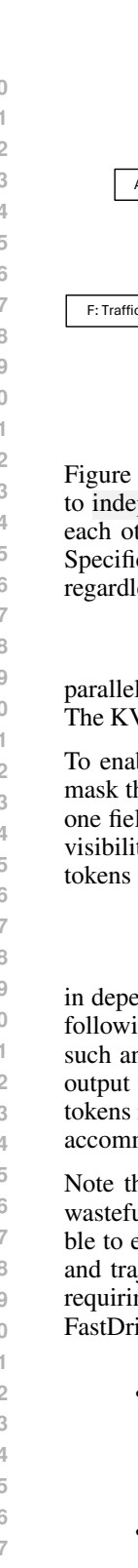

Figure 3: An example part of the dependency graph and the corresponding attention mask. Due to independency and thus possibility of parallelism, for example, A, B, D, and E cannot attend to each other. Other field pairs with dependency can have the latter attend to the former as usual. Specifically, the fixed tokens are pre-filled so they can be attended to by all subsequent tokens regardless of dependencies. Padding tokens cannot be attended to by other tokens.

parallel fields simultaneously in a single forward pass, all while maintaining the original batch size. The KV cache is thus fully shared and reused across all fields, maximizing computational efficiency.

To enable this parallel decoding strategy within a single sequence, we design a custom attention mask that enforces the correct causal dependencies. This mask prevents tokens being generated for one field from attending to tokens in other fields that are not guaranteed to be complete. Formally, visibility is determined from the dependency graph: a token in field $B$ is permitted to attend to tokens in field $A$ if and only if $A$ is an ancestor of $B$, i.e., there is a path

$$X_1 \to X_2 \to \cdots \to X_n; \quad X_1 = A \land X_n = B \land \forall i : (X_i, X_{i+1}) \in E(\mathcal{G}) \tag{1}$$

in dependency graph $\mathcal{G}$. As a specific case, the fixed tokens of the initial template are visible to all following tokens in the sequence, as they are known from the start. Figure 3 shows an example of such an attention mask. Since the generated fields have variable lengths, we pad or truncate their output to a pre-defined, fixed length. This ensures that the positional encodings for all subsequent tokens in the sequence can be determined correctly. The maximum length for each field is chosen to accommodate the outputs for at least 99% of the training data.

Note that a naive implementation would also process padding tokens, which is computationally wasteful. To avoid this overhead, we modify the attention mask to make all padding tokens invisible to every other token in the sequence, including those generated for the subsequent meta-action and trajectory. This strategy allows padding to reserve space and define token positions without requiring the padding tokens themselves to be processed during the inference forward pass. Overall, FastDriveCoT offers several key advantages regarding efficiency and implementation:

- **Zero Computational Overhead:** Our approach introduces no additional FLOPs compared to standard auto-regressive decoding. By managing all generation within a single sequence, the custom attention mask ensures that the KV cache is fully reused across all forward passes, eliminating any need to recompute existing key-value pairs.

- **Reduced Memory Bottleneck:** Packing tokens along the sequence length dimension allows the shared KV cache to be accessed by multiple query tokens (one for each parallel field) within a single CUDA attention kernel. This strategy reduces the total memory I/O of the primary bottleneck of LLM inference, thereby improving efficiency.

- **Implementation Simplicity:** The method is straightforward to implement. It achieves its efficiency gains through a careful and precise design of standard transformer components—namely the attention mask and position IDs—rather than requiring complex architectural modifications.

Table 1: Summary of Main Results

| Base Model | CoT | Meta Action (IOU) $\uparrow$[1] | Trajectory (ADE @ 3s) $\downarrow$ | Trajectory (ADE @ 6.4s) $\downarrow$ | CoT Time (s) $\downarrow$ | Overall Time (s) $\downarrow$ |
|---|---|---|---|---|---|---|
| Qwen2 0.5B + VLA-AR | None | 0.784 | 0.798 | 3.761 | - | 1.724 |
| | Autoregressive | **0.811** | 0.753 | 2.948 | 9.191 | 11.305 |
| | FastDriveCoT | 0.804 | **0.666** | **2.911** | 2.239 (4.1x)[2] | 4.723 (2.4x) |
| Qwen3 1.7B + VLA-AR | None | 0.799 | 0.717 | 3.470 | - | 2.569 |
| | Autoregressive | 0.801 | 0.686 | 2.863 | 13.186 | 16.033 |
| | FastDriveCoT | **0.815** | **0.639** | **2.686** | 4.189 (3.1x) | 8.246 (1.9x) |
| Qwen2.5-VL 3B + VLA-AR | None | **0.865** | 0.617 | 1.970 | - | 5.107 |
| | Autoregressive | 0.850 | 0.511 | 2.045 | 14.866 | 18.244 |
| | FastDriveCoT | 0.856 | **0.482** | **1.908** | 4.608 (3.2x) | 8.415 (2.2x) |
| Qwen2 0.5B + Transfusion | None | -[3] | 0.839 | 4.023 | - | 0.330 |
| | Autoregressive | - | **0.564** | **2.225** | 7.977 | 8.256 |
| | FastDriveCoT | - | 0.720 | 2.783 | 2.428 (3.6x) | 2.688 (3.1x) |

[1] $\uparrow$ indicates that higher is better; $\downarrow$ means that lower is better.
[2] Numbers in brackets indicate relative speedup compared to autoregressive CoT.
[3] We do not include meta-action in the sequence when using Transfusion.

# 4 EXPERIMENTS

## 4.1 EXPERIMENT SETTINGS

**Models.** We evaluate FastDriveCoT on three different base models with varying architectures and scales, Qwen2-0.5B (Qwen Team, 2024), Qwen3-1.7B (Yang et al., 2025a), and Qwen2.5-VL-3B (Bai et al., 2025). For Qwen2-0.5B and Qwen3-1.7B, which do not take vision inputs natively, we use Dinov2 (Oquab et al., 2023) to extract features from input frames and provide them as continuous input to the language model. We also encode 1.6 seconds of trajectory history into one embedding (via a small MLP) as an additional input to the LLM.

After the CoT output, we additionally produce meta-actions and future trajectories to evaluate the model's diving performance. We explore two types of architectures: (1) a purely autoregressive transformer (VLA-AR), where the CoT, meta-actions, and future trajectories are tokenized into discrete sequences, trained with a next-token prediction loss, and decoded autoregressively; and (2) Transfusion (Zhou et al., 2025a), in which the CoT and meta-actions are treated the same as in (1), but future trajectories are modeled via flow matching (Lipman et al., 2022), while sharing the same transformer backbone.

To represent trajectories in (1), we employ an auto-encoding pre-trained tokenizer that compresses each 6.4s future trajectory into 6 discrete tokens. For the representation of the trajectory in (2), we first convert the future trajectory into 64 $(\Delta x, \Delta y, \Delta \text{yaw})$ 10 Hz actions. These are then embedded using sinusoidal positional encodings followed by an MLP, producing 64 continuous embeddings. A lightweight MLP is used to decode the embeddings back into $(\Delta x, \Delta y, \Delta \text{yaw})$ action space.

To evaluate the effectiveness of our approach, we compare it against two baselines:

1. **No CoT:** A model trained end-to-end *without* intermediate CoT generation. This baseline measures the overall performance contribution of incorporating CoT.

2. **Autoregressive CoT:** A model that uses our full CoT template but generates the fields sequentially using standard autoregressive decoding. This baseline isolates and quantifies the efficiency gains specifically provided by our parallel decoding method.

**Data.** To evaluate the efficiency and task performance of FastDriveCoT, we leverage a large internal dataset[1] consisting of 20,000 hours of driving data from multiple ego-vehicles in 1700+ cities and 25 countries, which contains various road and weather conditions, day and night times, and different amounts of traffic. We use the trajectory data and synchronized recordings from the front- and back-view cameras, downsampled to a resolution of $320 \times 512$. We employ an auto-labeling pipeline containing Qwen2.5-VL-72B (Bai et al., 2025) to generate structured CoT data for our experiments.

---

[1] We will publicly release a large subset of this dataset upon publication (dataset name redacted for review).

For each data point, we randomly sample a timestamp and provide the model with the corresponding 2Hz front-view video and trajectory history as input. The model is then prompted to generate CoT data, with constrained decoding used to ensure the output strictly adheres to our pre-defined template. This pipeline yielded 717,344 high-quality training samples and 950 test samples.

**Evaluation metrics.** To validate FastDriveCoT's ability to improve computational efficiency while maintaining high task performance, we use the following evaluation metrics:

- **CoT Time:** The latency from when the model receives its inputs until CoT generation is complete. This metric directly measures the speedup of the core reasoning component targeted by FastDriveCoT.

- **Overall Time:** The total latency from model input to final trajectory output. This measures the end-to-end efficiency gain in a practical application.

- **Meta-Action IOU:** The model predicts a sequence of meta-actions over a 6.4-second horizon. We evaluate these predictions by calculating the Intersection over Union (IOU) between the predicted and ground-truth actions within each 0.1-second interval. This metric assesses the quality of the model's scene comprehension and high-level decision-making.

- **Trajectory ADE:** We measure the accuracy of the final motion plan using the Average Displacement Error (ADE) between the predicted and ground-truth trajectories.

**Training and inference configuration.** We initialize our models from their pre-trained checkpoints, adding randomly initialized embeddings and layers as required. The models are then trained for 50,000 steps on our dataset of 717,344 samples. We use a batch size of 64 with the AdamW optimizer (Loshchilov & Hutter, 2017) and a learning rate of $3 \times 10^{-5}$ with a cosine decay schedule.

All inference experiments are conducted on a single NVIDIA A100 80GB SXM GPU. We use BFloat16 precision for most calculations, with the exception of Float32 for processing the input trajectory history and the final output logits. Our implementation utilizes PyTorch's Scaled Dot Product Attention (SPDA), which dispatches to FlashAttention-2 (Dao, 2023) for standard causal masks (used in our baselines) and to xFormers (Lefaudeux et al., 2022) for the custom masks required by our parallel decoding method. For accurate latency measurements, we use the CUDA event timer and discard the first 10 iterations of each run to account for kernel warm-up effects.

## 4.2 MAIN RESULTS

**Efficiency results.** As summarized in Table 1, FastDriveCoT achieves a $3.1 - 4.1\times$ speedup in CoT generation time compared to a standard autoregressive baseline. This translates to a $1.9 - 3.1\times$ end-to-end speedup in overall inference time, depending on the proportion of time spent on subsequent meta-action and trajectory generation.

This significant acceleration makes the use of CoT more practical for real-world applications by mitigating the otherwise prohibitive latency of autoregressive reasoning. Furthermore, these performance gains are consistent across all tested model architectures and scales, demonstrating the general applicability of FastDriveCoT across a variety of VLMs and LLMs.

**Task performance.** Table 1 further shows that incorporating CoT provides substantial benefits for both meta-action and trajectory generation. The most significant gain is observed in the 3-second trajectory prediction with the Qwen2.5-VL 3B model, where ADE improves from 0.617 (No CoT baseline) to 0.511 with auto-regressive CoT, and further to 0.482 with our parallel decoding CoT. Strong improvements are also seen in longer-term (6.4 s) ADE across other models, such as the Qwen2 0.5B and Qwen3 1.7B, confirming the general effectiveness of CoT for AV tasks.

When comparing our parallel decoding method to the autoregressive CoT baseline, we find that FastDriveCoT maintains a highly competitive level of performance. In VLA-AR experiments, our parallel method performs slightly better, which we hypothesize is due to the structured decoding process inherently encouraging better adherence to the template format. In Transfusion experiments, while there is a minor degradation in trajectory ADE compared to the autoregressive CoT, FastDrive-CoT still significantly outperforms the baseline without CoT. In summary, FastDriveCoT delivers

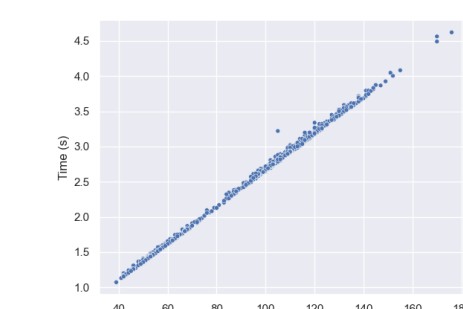 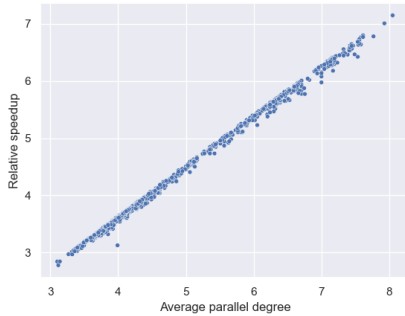

(a) CoT generation time and the number of tokens on the longest path.

(b) CoT generation relative speedup and average parallel degree.

Figure 4: Additional analysis on the generation time of each data sample in the test set.

significant improvements of computational efficiency while consistently preserving the substantial task accuracy improvements of CoT across all experiments.

### 4.3 INFERENCE ANALYSIS

To better understand the source of FastDriveCoT's speedup, we conduct a detailed analysis of the inference process using the Qwen2 0.5B + VLA-AR configuration.

Our analysis focuses on the critical path in the dependency graph: the longest chain of dependent tokens that must be generated sequentially. As shown in Figure 4a, we observe a strong linear relationship between the CoT generation time and the number of tokens on this critical path. This result confirms that the primary determinant of latency is the number of sequential forward *passes*, rather than the number of *tokens* generated in each forward pass or the total number of tokens generated across all fields. This finding opens two promising avenues for future work:

- Inference speed can be further improved by explicitly designing CoT templates to minimize the length of their critical dependency path.
- Since generating more tokens in parallel does not increase latency, FastDriveCoT could enable an agent to utilize a "fast response" trajectory (with little or no CoT) generated concurrently with a "comprehensive thinking" CoT at no additional time cost. We leave the application of such dual-response systems for future research.

Another key insight emerges from the relationship between the achieved speedup and the degree of parallelism, as illustrated in Figure 4b. Here, relative speedup is the ratio of the autoregressive baseline's CoT generation time to FastDriveCoT's, and the average parallel degree is the average number of tokens processed per forward pass. The plot reveals a strong linear relationship, confirming that the speedup is directly proportional to the degree of parallelism. We also observe that the speedup factor is consistently only slightly lower than the average parallel degree. It shows that the value of average parallel degree is approximately the speedup factor of FastDriveCoT in this scale. We hypothesize this slight difference is due to the less efficient attention kernel allowing the custom attention mask used in FastDriveCoT. We leave this further optimization for future research.

## 5 CONCLUSION

In this paper, we present FastDriveCoT: a method to accelerate template-structured CoT for AV tasks using parallel decoding. FastDriveCoT employs a generalizable dependency graph and an optimal dynamic programming algorithm to dynamically identify which fields can be generated in parallel. Experiments show that FastDriveCoT achieves a significant 3-4× improvement in CoT inference speed across diverse VLM/LLM architectures and scale, while preserving downstream task performance improvement on both meta-action accuracy and the quality of generated trajectories.

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

## A   THE USE OF LARGE LANGUAGE MODELS

During writing, we use LLMs to polish and rephrase certain paragraphs in the paper for better fluency. We do not use LLMs for research ideation.

