# OpenReview forum: "Accelerating Structured Chain-of-Thought in Autonomous Vehicles"
_ICLR.cc/2026/Conference — Submitted to ICLR 2026_

### Official Review · Reviewer_ZSwQ · 2025-10-29

**Soundness:** 3
**Presentation:** 2
**Contribution:** 3
**Rating:** 6
**Confidence:** 4

**Summary:**

This paper addresses the high latency of CoT reasoning in VLA models for autonomous driving. It proposes FastDriveCoT, a parallel decoding method that leverages a structured CoT template and dependency graph to generate multiple reasoning steps concurrently, achieving significant speedup without sacrificing task performance.

**Strengths:**

The paper tackles a critical and practical problem: the inference latency of CoT reasoning in time-sensitive autonomous driving applications . The quality of the proposed FastDriveCoT method is demonstrated through its well-structured approach, including the use of a CoT template, a dependency graph for parallelization, and an efficient implementation leveraging custom attention masks for shared KV caching . The clarity is good, explaining the parallel decoding mechanism well. Experimental results convincingly show substantial speedups while maintaining or even slightly improving downstream task performance.

**Weaknesses:**

The effectiveness of FastDriveCoT relies on a predefined, template-structured CoT. While suitable for the relatively standardized reasoning in AVs, this might limit flexibility or generalizability compared to free-form CoT. The approach also depends on high-quality, template-adherent CoT data, which was auto-labeled using a large VLM, potentially introducing noise or biases . While the paper claims zero computational overhead compared to autoregressive decoding, managing the dependency graph and custom attention masks might introduce implementation complexity. A minor performance degradation was observed in the Transfusion architecture experiments.

**Questions:**

1. How adaptable is FastDriveCoT to different CoT templates or reasoning structures? Does modifying the template (e.g., adding/removing fields, changing dependencies) require significant re-engineering of the dependency graph or attention masks?
2. The method relies on accurately following the template. How does the parallel decoding handle cases where the model deviates from the expected structure or produces errors within a field that might affect dependent fields generated later?
3. Could you elaborate on the comparison between FastDriveCoT and other LLM acceleration techniques like speculative decoding or lookahead decoding? Are these methods complementary, or does FastDriveCoT offer distinct advantages for structured CoT?
4. Why the Transfusion architecture show a slight degradation in trajectory ADE with FastDriveCoT compared to autoregressive method, while the VLA-AR architecture always showed improvement?

---

> ### Author Response · Authors · 2025-11-25
>
> Dear Reviewer ZSwQ,
>
> We sincerely thank you for your positive and constructive feedback.
>
> ### Responses to the Weakness
>
> > The effectiveness of FastDriveCoT relies on a predefined, template-structured CoT. While suitable for the relatively standardized reasoning in AVs, this might limit flexibility or generalizability compared to free-form CoT.
>
> We acknowledge that our method benefits from leveraging the relatively standardized reasoning structure common in AV tasks (in contrast to more abstract domains like mathematical reasoning).
> However, the proposed structure remains **flexible and easily adaptable** across different tasks. It **includes free-form CoT fields**, which make it well-suited even for more open-ended applications such as driving QA.
>
> > While the paper claims zero computational overhead compared to autoregressive decoding, managing the dependency graph and custom attention masks might introduce implementation complexity.
>
> Thank you for raising this point. The dependency graph and custom attention mask are handled on the CPU asynchronous with CUDA streams, ensuring minimal if not zero computational overhead during GPU execution.
> Most of inference acceleration techniques add additional complexity to the implementation, and we believe the 3–4$\times$ speedup observed in our experiments more than justifies this additional complexity.
>
> ### Answers to the Questions
>
> > How adaptable is FastDriveCoT to different CoT templates or reasoning structures? Does modifying the template require significant re-engineering of the dependency graph or attention masks?
>
> FastDriveCoT is designed to be **highly flexible and intuitive** to adapt. Modifying the structure only requires specifying task-relevant fields and their *direct* dependencies. The construction of the full dependency graph and the corresponding attention mask is **fully automated** by our algorithm, requiring no manual re-engineering.
>
> > The method relies on accurately following the template. How does the parallel decoding handle cases where the model deviates from the expected structure or produces errors within a field that might affect dependent fields generated later?
>
> The structured template prevents such deviations. Template tokens (mainly field titles) are prefilled during inference, and our algorithm ensures that the output format strictly adheres to the defined structure.
> As a result, LLM outputs cannot violate the template, and formatting errors within one field do not propagate to others due to isolated field decoding.
>
> > Could you elaborate on the comparison between FastDriveCoT and other LLM acceleration techniques like speculative decoding or lookahead decoding? Are these methods complementary, or does FastDriveCoT offer distinct advantages for structured CoT?
>
> These methods are complementary. Techniques such as speculative or lookahead decoding can be applied within each field of FastDriveCoT, providing stacked acceleration.
> While those approaches are general-purpose for autoregressive decoding, FastDriveCoT benefits from its structured format, achieving greater inherent speedup and remaining fully compatible with such techniques.
>
> > Why the Transfusion architecture show a slight degradation in trajectory ADE with FastDriveCoT compared to autoregressive method, while the VLA-AR architecture always showed improvement?
>
> We hypothesize that this minor degradation arises from increased optimization difficulty -- Transfusion combines two distinct loss functions, stacked with our specialized attention masks.
> Nonetheless, FastDriveCoT maintains a significant speedup and still outperforms the non-CoT baseline.
>
> We sincerely thank the reviewer again for their thoughtful comments, which have helped us better clarify the adaptability and practical efficiency of FastDriveCoT.

---

### Official Review · Reviewer_Vff3 · 2025-10-29

**Soundness:** 3
**Presentation:** 2
**Contribution:** 2
**Rating:** 2
**Confidence:** 3

**Summary:**

This paper introduces FastDriveCoT, a novel parallel decoding method to address the inference latency of Chain-of-Thought (CoT) reasoning in vision-language-action models for autonomous driving. FastDriveCoT decomposes the reasoning process into a dependency graph of distinct sub-tasks, allowing independent sub-tasks to be generated in parallel. By generating multiple independent reasoning steps concurrently in a single forward pass, the method significantly reduces sequential computations. Experiments show that FastDriveCoT achieves a 3-4x speedup in CoT generation and a substantial reduction in end-to-end latency across various model architectures, all while preserving the downstream task performance improvements brought by CoT.

**Strengths:**

- This paper addresses a very practical and significant problem. The robustness and interpretability of LLMs are crucial for AV systems, but their latency is a major obstacle to deployment.

- The method of decomposing CoT into a dependency graph and achieving parallel decoding in a single forward pass via a custom attention mask appears sensible. This approach can effectively utilize the parallel computing capabilities of modern GPUs and fully reuse the KV cache.

- The paper demonstrates significant speedup (3-4x in CoT time) without compromising on accuracy (i.e., not sacrificing downstream task performance).

**Weaknesses:**

- The core contribution heavily relies on a manually designed, highly fixed CoT template. Although the authors mention this template is an "example" (line 185), the entire methodology (including the dependency graph construction) is based on this fixed structure. Furthermore, when handling "multi-instance" fields (like lanes and critical objects), the method depends on a fixed number of slots (e.g., 3 time ranges for lanes, 4 critical objects). This might be fragile in complex, dynamic real-world scenarios.

-The experimental comparison is limited to "No CoT" and "Autoregressive CoT." The paper lacks comparisons with other SOTA LLM acceleration techniques. For instance, speculative decoding is a more general acceleration method that does not require a fixed template. Alternatively, could training a smaller model to mimic the CoT output achieve a similar balance between latency and performance? Without these comparisons, it is difficult to judge if FastDriveCoT is the best approach for this problem.

- The evaluation relies entirely on a large "internal dataset." The authors do not report results on any public, standard autonomous driving benchmarks (like nuScenes or Waymo Open Dataset). This severely harms the work's reproducibility. Although the authors promise to release a subset of the dataset (line 376), during the review period, we cannot independently verify the findings or fairly compare this method with other SOTA methods trained on public benchmarks.

**Questions:**

1. Have the authors considered a comparison with Speculative Decoding? It seems to be a more general acceleration technique that does not require a predefined CoT structure.

2. How does the method's performance (both speed and accuracy) degrade when the number of entities in a real-world scene (e.g., critical objects) exceeds the fixed slots in the template?

---

> ### Author Response · Authors · 2025-11-25
>
> Dear Reviewer Vff3,
>
> We appreciate your time and constructive feedback on our paper.
>
> ### Responses to Weakness
>
> > relies on a manually designed, highly fixed CoT template.
>
> We would like to clarify that the template design is **flexible and easily adaptable** to different tasks. It can be modified with minimal effort to accommodate task-specific requirements while preserving the same underlying reasoning flow.
>
> > the entire methodology (including the dependency graph construction) is based on this fixed structure
>
> While our methodology operates on a structured template, **the structure itself** is not fixed. It can be readily transferred or modified for other tasks, as its definition is intuitive and task-driven. Moreover, both the dependency graph and the attention mask are **automatically derived** from the direct dependencies by our algorithm, rather than being manually specified or tied to a particular format or reasoning structure.
>
> > the method depends on a fixed number of slots
>
> The number of slots is empirically chosen to cover at least 99% of the dataset. It can be easily adjusted for different domains or dataset characteristics without changing the core methodology. Besides, since the slots are generated in parallel during inference, the inference time of our method does not grow with the number of slots, achieving more speedup with more slots.
>
> > lacks comparisons with other SOTA LLM acceleration techniques
>
> We thank the reviewer for pointing this out and for mentioning speculative decoding as an example.
> Our approach is orthogonal to such acceleration methods and can be **combined** with many of them. For instance, speculative decoding can be applied independently to each field in our structured reasoning process.
> Additionally, techniques like speculative decoding are typically constrained by the accuracy of lookahead tokens, leading to only 1.5-2.5$\times$ acceleration \[1-2\], whereas our method achieves substantially greater efficiency.
>
> > could training a smaller model to mimic the CoT output achieve a similar balance between latency and performance
>
> This line of idea is also orthogonal to our methods. A smaller model trained to mimic CoT reasoning can still benefit from our structured inference mechanism, with stacked acceleration when combined.
>
> > the evaluation relies entirely on a large "internal dataset."
>
> We regret that the dataset cannot be released due to internal policies, and we did not evaluate on public datasets for the same reason. Nevertheless, our method primarily demonstrates speedup, rather than performance improvement. This efficiency gain is clear to be **independent of specific dataset** and thus expected to transfer to other datasets or benchmarks.

---

> ### Author Response · Authors · 2025-11-25
>
> (continued)
>
> ### Answers to the Questions
>
> > Have the authors considered a comparison with Speculative Decoding? It seems to be a more general acceleration technique that does not require a predefined CoT structure.
>
> We did not have sufficient time during the response period to conduct this additional experiment. However, existing studies report that speculative decoding typically achieves 1.5–2.5$\times$ speedup \[1-2\], which is notably lower than the acceleration achieved by our method.
> Moreover, our approach is combinable with speculative decoding (applied on each field), stacking the acceleration.
>
> > How does the method's performance (both speed and accuracy) degrade when the number of entities in a real-world scene (e.g., critical objects) exceeds the fixed slots in the template?
>
> The speed will not be impacted when exceeding the fixed slots. The inference time does not grow with more slots since the slots are decoded in parallel, thus our method can potentially have more speedup with more slots.
>
> \[1\] Fu, Y., Bailis, P., Stoica, I., & Zhang, H. (2024). Break the sequential dependency of llm inference using lookahead decoding. *arXiv preprint arXiv:2402.02057.*
>
> \[2\] Leviathan, Y., Kalman, M., & Matias, Y. (2023, July). Fast inference from transformers via speculative decoding. *In International Conference on Machine Learning (pp. 19274-19286).* PMLR.
>
> ### Summary
>
> In summary, our reasoning template and dependency graph design are **flexible, intuitive, and easily generalizable** across different tasks. The automation of graph and attention mask construction ensures minimal manual overhead, while **the speedup achieved is dataset-independent** and can **complement other acceleration methods**.
>
> We sincerely thank the reviewer for their insightful comments, which have helped us better clarify the scope and generality of our approach.

---

> > ### Comment · Reviewer_Vff3 · 2025-11-28
> >
> > Thank you for your response. However, to be honest, it has not dispelled my doubts.
> >
> > I share the similar concerns raised by Reviewer j95Y and ZSwQ regarding the generalization of the fixed template slots and the exclusive use of an internal dataset. I firmly believe that evaluating the method on open benchmarks is necessary to properly assess its validity and reproducibility.
> >
> > Therefore, I will maintain my original rating.

---

### Official Review · Reviewer_HLxv · 2025-11-01

**Soundness:** 3
**Presentation:** 3
**Contribution:** 3
**Rating:** 6
**Confidence:** 2

**Summary:**

The paper proposes FastDriveCoT, a parallel decoding method that accelerates template-structured CoT. The paper targets the problem that normal chain of thought for driving is too slow for 10 Hz control because autoregressive decoding is fully sequential. Experiments like Table 1 shows FastDriveCoT gives 3.1$\times$ to 4.1$\times$ CoT speedup while keeping or improving meta action IOU and ADE over autoregressive CoT.

**Strengths:**

- The reviewer found the proposed idea to predefine a CoT template and decode independent fields in parallel using a dependency graph to be interesting.
- Again, interestingly parallel decoding even slightly improves template adherence so trajectory ADE at 3 seconds for Qwen2.5 VL 3B improves to 0.482 from 0.511 showing structure can help quality.
- Experiments in Table 1, the ablation style analysis in Figure 4, show consistent 3$\times$ to 4$\times$ CoT speedup with only small drops in some long horizon ADE.
- Writing is clear.

**Weaknesses:**

- Table 1 only compares no CoT and standard autoregressive CoT but it should also compare against shorter skeleton of thought decoding or speculative decoding baselines which are natural for speed claims.
- Some typos the reviewer could see: Line 115: dependecies -> dependencies; Figure 3 caption independency -> independence; Line 352 diving -> driving
- See questions below.

**Questions:**

- In Table 1 the Qwen2 0.5B + VLA-AR row shows meta action IOU 0.811 for autoregressive CoT but FastDriveCoT is 0.804 while trajectory ADE improves. Can you explain why meta action degrades when both use the same template.
- Figure 4 reports that speedup is almost linear in average parallel degree but Table 1 shows overall time speedup is only 1.9x to 2.4x for some models. Please clarify what overhead outside CoT causes this gap.

---

> ### Author Response · Authors · 2025-11-25
>
> Dear Reviewer HLxv,
>
> We sincerely thank you for your thoughtful and encouraging feedback.
>
> ### Responses to Weaknesses
>
> Thank you for pointing out the typos — we will correct them in the revised version.
>
> > Table 1 only compares no CoT and standard autoregressive CoT but it should also compare against shorter skeleton of thought decoding or speculative decoding baselines which are natural for speed claims.
>
> We appreciate this suggestion. Unfortunately, we did not have sufficient time during the response period to include speculative decoding experiments. However, prior work reports that speculative decoding typically achieves 1.5–2.5$\times$ speedup \[1-2\], which is considerably lower than the acceleration achieved by FastDriveCoT.
> Importantly, our method is complementary to speculative decoding and **can be combined** with it by applying speculative decoding independently on each field—leading to stacked acceleration.
>
> We also acknowledge that shorter skeleton-of-thought reasoning may offer a different trade-off between performance and efficiency. However, both the autoregressive and FastDriveCoT experiments were conducted using the same annotated dataset, ensuring a fair and controlled comparison.
>
> ### Answers to Questions
>
> > In Table 1 the Qwen2 0.5B + VLA-AR row shows meta action IOU 0.811 for autoregressive CoT but FastDriveCoT is 0.804 while trajectory ADE improves. Can you explain why meta action degrades when both use the same template.
>
> We hypothesize that this small difference stems from variance in the accuracy of specific reasoning segments within the CoT trace.
> Nonetheless, the deviation is within 1%, which falls in the range of normal statistical variation.
>
> > Figure 4 reports that speedup is almost linear in average parallel degree but Table 1 shows overall time speedup is only 1.9x to 2.4x for some models. Please clarify what overhead outside CoT causes this gap.
>
> For the VLA-AR models, the output adds 64 action tokens where we did not apply special accleration techniques, resulting in a lower overall speedup.
> In contrast, the Transfusion model uses only 10 denoising steps, which constitute a smaller portion of total inference time and therefore exhibit higher overall acceleration.
> The speedup reported in the CoT portion of Table 1 still directly reflects the core efficiency gains introduced by FastDriveCoT.
>
> We again thank the reviewer for the insightful comments and constructive feedback, which have helped us clarify our results and strengthen the presentation of our work.

---

### Official Review · Reviewer_j95Y · 2025-11-01

**Soundness:** 3
**Presentation:** 2
**Contribution:** 2
**Rating:** 2
**Confidence:** 4

**Summary:**

In this paper, the authors are presenting "FastDriveCoT". This is a  parallel decoding method that accelerates Chai of Thought reanosing for decision making. Here the authors apply this to the field of autonomous driivng. Overall, their approach is achieving a speed up of CoT reaosning and therefore reduces the overall end-to-end latenc in the AV software stack.

**Strengths:**

- Good overall novelty for AV: The presented approach shows good novelty in the field of AV reasoning.
- The authors present good technical innovation, by combining the structure CoT tmeplate, the dynamic programming algorithm and by maintining the zero extra FLOPs
- Good emperical results by showing the speed up of 3-4 times CoT reasonsing and therfore 2x faster E2E inference
- Good emperical results with different VLMs like Qwen2, Qwen2.5 and Qwen3
- Good comprehensive albation studies for effieciency and performance trade-offs

**Weaknesses:**

Currently, the paper shows a very good technical approach, but there are strong weaknesses that questions the papers overall impact:

- Currently, the method relies on highly structured reasoning templates that are selected by the authors for the driving task. It is unclear to the reader, if this enalbes generalization at all. My concern here is that this approach will not generalize very well to open-ended reasoning tasks. This is unfortunately something we see in AV every day.
- This dependence on predefined templates requires very specific CoT fields, dependency graphs, and manual  definitions. THis is an enginnering overhead and manual labor, that will not impact the AV world especially when we have other approaches that scale better
- Althought he authors show it can be conceptionally efficient, the custom attention maks highly overcomplicate the integration for standard frameworks
- The biggest flaws is the comparison of the approach. ALl experiments run on an internal dataset, an external validation or open benchmark on CARLA or other open datasets is not given

**Questions:**

1. How flexible is the COT Template design you have choosend?
2. Is the dependacy graph currently hand-crafted by you or is it learned from data?
3. How does your approach scale in multi-GPU setups?
4. How does your approach work on embedded hardware e.g. nvidia jetson so it can be implemented in a car?
5. Have you evaluated your approach on open benchmakrs or other simulations?

---

> ### Author Response · Authors · 2025-11-25
>
> Dear Reviewer j95Y,
>
> We appreciate your constructive comments on this paper and the strength points you have mentioned. Here we clarify several points in response to the weaknesses and your problems.
>
> ### Responses to the Weaknesses:
>
> > the method relies on highly structured reasoning templates; this approach will not generalized very well to open-ended reasoning task
>
> We clarify that a major part of the structural template invovles factual details necessary for the reasoning, necessary and generalizable to any reasoning task using CoT. It also includes open-ended summaries and reasoning for each aspect of reasoning in AV tasks, as well as an open-ended overall reasoning result. We have also shown in the experiment that this method contributes to both meta actions and trajectory predictions.
> We note that these two outputs are most common evaluation metric in an end-to-end AV model, and open-ended overall reasoning can additionally generalize to other AV research problems such as driving QA.
>
> > predefined templates requires very specific CoT fields, dependency graphs, and manual definitions; an engineering overhead and manual labor
>
> Defining the CoT fields and direct dependencies is intuitive, as we only need to specify the reasoning relevant elements and *direct* dependencies in the graph—Algorithm 1 automatically handles chained dependencies during inference.
> Moreover, most fields are standard and widely shared across AV tasks. Only minimal modifications are required for task-specific inputs or outputs (e.g., adding fields for optional sensors).
> Overall, the required effort is substantially lower than typical prompt engineering and less than the corresponding input pre-processing and output post-processing, given the intuitiveness of the design.
>
> > the custom attention mask highly overcomplicate the integration for standard framework
>
> It is important to note that all experiments use the simplest implementation of `torch.nn.functional.scaled_dot_product_attention`, where the attention mask is automatically computed by our algorithm.
> Therefore, the integration overhead is minimal to negligible.
> Moreover, we note that the comparison with baselines in the paper already accounts for a potential speed disadvantage from using this straightforward implementation instead of state-of-the-art methods like FlashAttention.
> Furthermore, since our attention mask is sparser than the standard causal mask, optimized attention kernels could potentially further accelerate our approach.
>
> > all experiments run on an internal dataset
>
> While we are unable to release the dataset due to internal policies, we highlight that our method primarily demonstrates **speedup**, not performance improvement.
> This speedup property is independent of dataset specifics and thus clearly transferable to other datasets.

---

> ### Author Response · Authors · 2025-11-25
>
> (continued)
>
> ### Answers to the Questions:
>
> > How flexible is the CoT Template design you have chosen?
>
> As described in the paper, the template design is highly flexible. For example, we support length-varying (open-ended) fields, arbitary dependecy (graph) structure, and employ an enumeration-then-elaboration scheme to handle multiple instances in parallel.
>
> > Is the dependacy graph currently hand-crafted by you or is it learned from data?
>
> The direct dependencies are manually specified, which is straightforward and requires minimal effort. For instance, summarizing traffic rules naturally depends on prior recognition of traffic signs and lights. Since these dependencies are intuitive, we do not need and did not conduct ablations on them. The overall dependency graph composition and handling of chained dependencies are fully automated.
>
> > How does your approach scale in multi-GPU setups?
>
> Our method is orthogonal to multi-GPU acceleration techniques and can fully leverage existing parallelization methods such as data, pipeline, tensor, and context parallelism.
> We can stack the similar speedup on the computation over those setups.
> Additionally, because our attention mask is sparser than the standard context mask, inter-GPU communication for context parallelism can potentially be reduced.
>
> > How does your approach work on embedded hardware e.g. nvidia jetson so it can be implemented in a car?
>
> Our approach is independent of the specific hardware as long as they are modern CUDA-based architectures, including NVIDIA Jetson.
> We have not tested on other hardware backends, as they are rarely used and we currently lack access to them.
>
> > Have you evaluated your approach on open benchmakrs or other simulations?
>
> We have not, due to internal restrictions. However, as noted above, the demonstrated speedup is expected to transfer across datasets and benchmarks.
>
> ### Summary
>
> We sincerely thank the reviewers for their thoughtful feedback and constructive suggestions, which have helped us clarify and strengthen the presentation of our work.
>
> In summary, our method provides a **flexible and intuitive** reasoning template design with **easily generalizable** template and dependency graph. The CoT fields and direct dependencies are straightforward to define and can generalize naturally across AV tasks, enabling effortless adaptation to new settings with minimal manual effort.
>
> Moreover, the demonstrated **speedup** effect is **inherently independent of specific datasets**, as it arises from architectural and algorithmic improvements rather than dataset characteristics. This indicates strong potential for transferring our approach to various benchmarks and applications without performance degradation.

---

> > ### Comment · Reviewer_j95Y · 2025-11-27
> >
> > Dear authors, thank you very much for the feedback. Since this is more of a clarification from your side than an enhancement of the paper and is missing the context for the real-world hardware, the open evaluation on other benchmarks or simulations, i do not see an improvement of the paper and therefore stick to my recommendation.

---

### Meta-Review · Area_Chair_j8X7 · 2026-01-09

**Summary:**

This paper was reviewed by four experts in the field, resulting in recommendation scores of (2, 2, 6, 6). The consensus among reviewers is that the paper is not currently ready for publication and requires significant revision, particularly regarding experimental evaluation, presentation, and comparative analysis. Reviewer j95Y raised concerns about the generalization ability of the proposed approach, suggesting that further evaluation on external datasets and benchmarks is needed to validate its effectiveness. Reviewer HLxv expressed concerns regarding the experimental comparisons and the overall presentation of the paper. Reviewer Vff3 echoed similar concerns, noting that the proposed solution and design appear complex and that its generalization ability may be limited without further experimental validation. Reviewer ZSwQ requested more detailed comparisons and in-depth analysis. The manuscript requires further polishing to make the motivation clearer and more convincing. Based on these concerns, the paper cannot be accepted at this time. The authors are encouraged to incorporate the reviewers' comments when revising the paper for submission elsewhere.

**Reviewer Concerns:**

Some of the missing discussion, experimental comparisons, and paper presentations are addressed by the rebuttal. While some more detailed experimental evaluations and in-depth analysis are still outstanding and are not convincing.

**Reviewer Scores:**

The reviewers would be satisfied with the author's commitment to adding missing discussion and polishing the paper presentation in the revised version, while they may still hold concerns regarding the provided experimental evaluations and in-depth analysis.

---

### Decision · Program_Chairs · 2026-01-26

Reject